# The Effects of Engeletin on Insulin Resistance Induced in Human HepG2 Liver Cells

**DOI:** 10.3390/cimb47070535

**Published:** 2025-07-10

**Authors:** Erdem Toktay, Secil Nazife Parlak, Tugba Kavas, Harun Un, Rustem Anıl Ugan, Muhammed Yayla

**Affiliations:** 1Department of Histology and Embryology, Faculty of Medicine, Kafkas University, Kars 36000, Turkey; erdemtoktay@gmail.com; 2Department of Histology and Embryology, Faculty of Medicine, Agri Ibrahim Cecen University, Agri 04100, Turkey; snparlak@agri.edu.tr; 3Department of Medical Histology and Embryology, Faculty of Medicine, Kafkas University, Kars 36000, Turkey; 4Department of Biochemistry, Faculty of Pharmacy, Agri Ibrahim Cecen University, Agri 04100, Turkey; hun@agri.edu.tr; 5Department of Pharmacology, Faculty of Pharmacy, Ataturk University, Erzurum 25000, Turkey; anil.ugan@atauni.edu.tr; 6Department of Pharmacology, Faculty of Medicine, Selcuk University, Konya 42000, Turkey; muhammed.yayla@gmail.com

**Keywords:** insulin resistance, ISR-1, ISR-2, GLUT-2, oxidative stress, flavonoids, Engeletin

## Abstract

In this study, we aimed to investigate the effect of Engeletin (ENG) on insulin resistance and the associated oxidative cell damage in human HepG2 liver cells. The cells were grown in a cell culture medium, and insulin resistance was induced. After the determination of the toxic and effective doses of Engeletin, the effects of Engeletin on insulin resistance and insulin resistance-induced oxidative damage, inflammation, and apoptosis. To induce IR, culture plates were treated with 30 mM glucose and 50 nM insulin and incubated for 48 h. Engeletin and metformin were given one hour before starting the insulin resistance induction. In the HepG2 cells, insulin resistance decreased glucose consumption, the expression of ISR-1 and ISR-2, and the GLUT-2 levels, while they were all increased by Engeletin, which showed a metformin-like effect. In addition, Engeletin alleviated oxidative cell damage by decreasing MDA levels, which increased due to insulin resistance-induced oxidative stress, increasing the GSH and SOD levels and decreasing the caspase-3 (Cas-3), caspase-9 (Cas-9), and tumor necrosis factor alpha (TNF-α) levels, which also increase under insulin resistance conditions. Engeletin was found to have the protective and therapeutic effect of reducing insulin resistance (IR) and the oxidative cell damage it causes in human HepG2 cells.

## 1. Introduction

Today, unhealthy and unbalanced nutrition, obesity, sedentary lifestyles, aging, and urbanization are increasing the number of patients with diabetes and dyslipidemia day by day, so obesity-related diseases have now become endemic. Unfortunately, the number of people living with obesity-related diabetes and diabetes-related complications is growing every year. More than half a billion people worldwide suffer from diabetes, which corresponds to more than 10.5% of the world’s adult population. According to the available data, the global prevalence of diabetes in people aged 20 to 79 years is estimated to be 536.6 million, which is expected to rise to 783.2 million in 2045. At the same time, the cost burden of increasing diabetes cases is estimated to exceed USD one trillion by 2045 [1]. This widespread prevalence of diabetes in the population will not only increase the cost burden but also cause the occurrence of serious complications related to diabetes [2].

Being the main factor regulating blood sugar, insulin is closely related to metabolic diseases and may be a risk factor in the emergence of some chronic diseases. Some of the metabolic diseases associated with insulin resistance (IR) include diabetes, tumors, cardiovascular and cerebrovascular diseases, and polycystic ovary syndrome. Increased insulin resistance is the source of many metabolic diseases, so determining the etiology, pathogenesis, and current treatment strategies will be important in the context of the disease [3]. Reducing IR and delaying the emergence of diabetes and its complications has become a focus in related fields. Researchers have investigated many drug therapies, including flavonoid groups rich in biological activity, to achieve this goal [4]. Flavonoids are effective compounds against insulin resistance with proven medicinal value. Numerous studies support the potential benefits of flavonoids in IR and diabetes [5]. Engeletin (ENG), a flavonoid group, belongs to the lily family and is one of the phytochemicals found naturally in plants [6]. This compound is a flavonoid glycoside obtained from different species, especially wine, Hymenaea martiana, Petiveria alliacea, and Engelhardia roxburghiana. Recent studies have shown that Engeletin has a strong anti-inflammatory effect [7,8]. Huang et al. showed in their study that Engeletin is a powerful antioxidant and protects neuron cells against oxidative stress [9]. Researchers have also suggested that Engeletin may have anti-cancer effects [10]. Related to our subject, there are studies on the anti-diabetic effects of plants containing Engeletin [11,12]. There are also studies indicating that Engeletin may have an effect on aldose reductase inhibition and the formation of glucose-dependent advanced glycation products [13]. Engeletin does not directly stimulate insulin secretion but regulates blood glucose levels. However, studies on the activity of Engeletin on adipocytes have shown that it may stimulate brown adipose tissue by activating mitochondria [6]. Therefore, we think that Engeletin may be effective in insulin resistance due to diseases such as obesity and diabetes.

In light of all this information, the possible effects of Engeletin, the active substance we use in this study, may be effective in terms of reducing both insulin resistance and the oxidative stress damage it causes. Therefore, in our study, we aimed to investigate the effects of Engeletin in an insulin resistance model established in the HepG2 cell line.

## 2. Materials and Methods

### 2.1. Cell Line Used in Experiments

The human HepG2 cell line used in our studies was obtained from the ATCC. HepG2 cells were cultured in the Central Laboratory of Kafkas University through routine passaging once a week. Cells were grown in heat-inactivated 20% fetal bovine serum (FBS; Sigma-F7524, New York, NY, USA) and antibiotics (100 units/mL penicillin G and 100 μg/mL streptomycin, Gibco-15240-122, St. Louis, MO, USA) in RPMI 1640 (Gibco-31800-014, St. Louis, MO, USA) medium at 37 °C in a humidified incubator containing 5% CO_2_ and 95% air under 1-atmosphere pressure. The cells grown in sterile culture dishes were used for the experiments when they covered 60–80% of the culture plate surface.

### 2.2. Preparation of Drugs Used in Experiments and Determination of Cytotoxicity with MTT Assay

Engeletin was obtained from the supplier company (Medchemexpress company, Engeletin: HY-N0436, Monmouth Junction, NJ, USA). After it was dissolved with 0.1% dimethyl sulfoxide (DMSO), its cytotoxic effects were studied at the concentrations of 100 µM, 10 µM, 1 µM, 100 nM, 10 nM, 1 nM, and 100 pM.

When the cells used in the experiments covered the culture plates by 60 to 80%, they were removed from the surface of the culture plate to which they adhered with trypsin. The total cell count was calculated using a hemocytometer. Five thousand HepG2 cells from 100% viable single-cell suspensions were seeded into each of the 96-well culture dishes. The seeded cells were incubated at 37 °C for 24 h to allow them to adhere to the surface of the culture plate and proliferate. After 24 h of incubation, the cells were treated with Engeletin at different concentrations, and the effects on cell viability were determined with the MTT assay at 24, 48, and 72 h.

### 2.3. Induction of Insulin Resistance in Cell Culture

We used the following modifications in accordance with the literature to develop a model of insulin resistance in the HepG2 cells [14,15]: Twenty thousand HepG2 cells from 100% viable single-cell suspensions were seeded into each of the 24-well culture dishes. When the cells used in the experiments covered the culture plates, the medium was removed from the culture plate. Then, 30 mM glucose and 50 nM insulin were administered to the culture plates to induce IR. The cells were incubated at 37 °C for 48 h to allow them to develop IR. Engeletin and metformin were given one hour before starting the insulin resistance induction. After 48 h of incubation, the cell media were discarded, and only glucose at concentrations of 5.5 mM or 16 mM was added to all groups. Then, following 50 min of incubation, 100 nM insulin was added to all groups to induce glucose consumption and incubated for 10 min. Afterwards, the amount of glucose in the medium was measured.

### 2.4. Experimental Groups

The experimental groups in our study are as follows:1.Control (HEALTHY GROUP);2.High-dose glucose + high-dose insulin (IR GROUP);3.IR + Engeletin at 100 µM (IR + ENG100 GROUP);4.IR + Engeletin at 10 µM (IR + ENG10 GROUP);5.IR + Metformin (control drug) (IR + MET GROUP);6.IR + Metformin + Engeletin at 100 µM (IR + MET+ENG100 GROUP);7.IR + Metformin + Engeletin at 10 µM (IR + MET + ENG10 GROUP).

Then, the Insulin-Dependent Glucose Consumption Test was performed again to test the effect of Engeletin, and glucose consumption was determined. Subsequently, the cells were harvested from the flask with the scraping method and stored for molecular and biochemical analyses.

### 2.5. Molecular Analyses

#### 2.5.1. mRNA Isolation

The cells were homogenized with Tissue Lyser II (Qiagen, Hannover, Germany). RNA extraction was performed. The total RNA isolation steps were carried out using the Qiaqube RNA isolation kit (Hilden, Germany)as recommended by the manufacturer. The total mRNA was obtained and then purified using the RNeasy Mini Kit on a QIACUBE (Qiagen, Hilden, Germany) device according to the manufacturer’s instructions. The total mRNA amount was measured with nano-drop spectrophotometry (EPOCH Take3 Plate, Biotek, Shoreline, WA, USA) at 260 nm.

#### 2.5.2. Reverse Transcriptase Reaction and cDNA Synthesis

The High Capacity cDNA Reverse Transcription Kit (Waltham, MA, USA) enzyme was used to synthesize cDNA from total RNA. Each reaction was performed with 10 μL of RNA, and cDNA synthesis was carried out using the Veriti 96-Well Thermal Cycler (Applied Biosystems, Waltham, MA, USA) at 25 °C for 10 min, 37 °C for 120 min, and 85 °C for 5 min. The resulting cDNA was stored at −20 °C.

#### 2.5.3. Quantitative Determination of mRNA Expression

Tumor necrosis factor alpha (TNF-α), insulin receptor substrate 1 (ISR-1), insulin receptor substrate 2 (ISR-2), glucose transporter 2 (GLUT-2), caspase-3 (Cas-3), and caspase-9 (Cas-9) mRNA expression was quantified using the Gene Expression Master Mix kit (Austin, TX, USA). Amplification and quantification were performed on the StepOne Plus Real-Time PCR System (Applied Biosystems). For 10 ng of cDNA, SYBR Green was pipetted according to the amounts recommended by the company, and 40 cycles of PCR were performed. mRNA expression was quantified using the instructions of the Applied biosystems PowerUp SYBR Green Master Mix kit (Austin, TX, USA). For 10 ng of cDNA, 10 µL of PowerUp SYBR Green Master Mix (2X) and 10 µL of forward and reverse primers + DNA template + Nuclease-Free Water were pipetted, for a total volume of 20 µL, according to the SYBR Green Master Mix (Austin, TX, USA) user guide, and 40 cycles of PCR were performed according to the following temperature values: if primary Tm < 60 °C, standard cycling mode was used for 2 min at 50 °C, for 2 min at 95 °C, for 15 s at 95 °C, for 15 s at 55 °C, and for 1 min at 72 °C).

The results were subtracted from the reference gene beta-actin value, and the Ct values were automatically converted to delta delta Ct on the device. All data are expressed as fold changes in expression compared with the control (healthy) group by using the 2^−ΔΔCt^ method [16].

The results obtained are statistically and graphically presented. The primer sequences used for the real-time PCR tests are provided in Table 1.

### 2.6. Biochemical Analyses

#### Oxidative Stress Analysis

Oxidant and antioxidant parameters—malondialdehyde (MDA), glutathione (GSH), and superoxide dismutase (SOD) levels—were measured using human-specific ELISA kits (Wuhan, China) (MDA: ELABSCIENCE (E-BC-K025-M); GSH: ELABSCIENCE(E-EL-H5410); and SOD: ELABSCIENCE (E-EL-H1113)) with two repetitions. Optical densitometer values were calculated in the ELISA reader following the procedures specified in the kit procedures. The protein amounts in the supernatants were measured manually using the Lowry method [17,18,19]. The mean absorbance of each sample and standard was calculated. All data are shown as means ± standard deviations (SDs) relative to each mg of protein.

### 2.7. Statistical Analyses

The results obtained were statistically evaluated with the IBM SPSS 22 package program. The GraphPad Prism 5.01 application was used to draw the graphs. The data obtained from our study were first tested for normal distribution. According to the Kolmogorov–Smirnov tests, it was determined that our data fit the normal distribution. Since there were more than 2 independent groups in our study, a one-way ANOVA test was used to compare variables, and Tukey’s test was used as a post hoc analysis. The mean and standard deviation values of the data were used. *p* < 0.05 was considered statistically significant.

## 3. Results

### 3.1. Findings on Effective Dose of Engeletin and Glucose Consumption

The MTT results, shown below in the figure, reveal that Engeletin was found to be non-toxic at all doses at 24, 48, and 72 h (Figure 1A–C).

Therefore, it was decided to use the two highest doses (100 µM and 10 µM) in the insulin resistance model, since it was thought that the efficacy of Engeletin would decrease depending on the dose.

Engeletin at these two doses was administered to HepG2 cells cultured in 24-well plates with two replicates. At 48 h, the media were collected, and the sugar contained in them was measured. The results were calculated as the remaining glucose in each well, as represented graphically in Figure 2A. Glucose consumption was significantly decreased in the groups with IR compared with the healthy group (*p* < 0.05). In the Engeletin and metformin groups, glucose consumption increased significantly (*p* < 0.05). No significant increase was observed in the combination groups compared with those with metformin and Engeletin alone (*p* > 0.05).

Following these studies, molecular and biochemical analyses were performed to elucidate the mechanism of the effects of Engeletin. In these groups, TNF-α, ISR-1, ISR-2, GLUT-2, Cas-3, and Cas-9 mRNA expression was measured with real-time PCR, and MDA, GSH, and SOD levels were measured with an ELISA reader and compared between the groups.

### 3.2. Real-Time PCR Results

#### 3.2.1. ISR-1 mRNA Expression Results

According to the analyses, ISR-1 mRNA expression was suppressed in the IR group compared with the healthy group. However, metformin administration, which is also used in routine treatment, increased the expression level of ISR-1. Similar expression increases were observed in the group administered with Engeletin. Furthermore, the co-administration of metformin and Engeletin had a synergistic effect and increased the expression level even more (Figure 2B). This result shows that Engeletin increased the expression level of ISR-1 at a dose of 100 µM and showed activity against insulin resistance.

#### 3.2.2. ISR-2 mRNA Expression Results

According to the analyses, ISR-2 mRNA expression was suppressed in the IR group compared with the healthy group. On the other hand, metformin administration, which is also used in routine treatment, increased the expression level of ISR-2. Similar expression increases were observed in the group administered Engeletin. However, the co-administration of metformin and Engeletin brought the ISR-2 level closer to the healthy group’s level (Figure 2C). This result indicates that, at a dose of 100 µM, Engeletin showed activity against insulin resistance by increasing the expression of ISR-2, showing a metformin-like effect.

#### 3.2.3. GLUT-2 mRNA Expression Results

According to the molecular analyses, GLUT-2 mRNA expression was significantly decreased in the IR group compared with the healthy group. On the other hand, metformin administration, which is also used in routine treatment, increased the GLUT-2 level compared with the IR group and brought it closer to the level found in the healthy group. In the group administered with Engeletin, a similar increase in expression was observed compared to the IR group. The co-administration of metformin and Engeletin had a synergistic effect and brought GLUT-2 expression closer to the healthy group’s level (Figure 2D). These results reveal that, especially at a dose of 100 µM, Engeletin showed efficacy against insulin resistance by increasing the expression of GLUT-2, showing a metformin-like effect.

#### 3.2.4. Cas-3 mRNA Expression Results

An important parameter to consider when examining the damage caused by insulin resistance in the cell is the level of caspase-3. Here, caspase-3 level showed high expression in the IR group compared with the healthy group. In the metformin-treated group, this level decreased compared with the IR group. On the other hand, Engeletin administration significantly decreased the Cas-3 levels, bringing them closer to those observed in the healthy group, while there was no significant difference between the IR + MET + ENG100 group and the healthy group (Figure 3A). These results reveal that Engeletin reduces IR-induced oxidative damage.

#### 3.2.5. Cas-9 mRNA Expression Results

Another important parameter when evaluating the damage caused by insulin resistance in the cell is the level of caspase-9, one of the inducers of caspase-3. When the expression levels of this molecule were examined, it was observed that it showed high expression in the IR group compared with the healthy group. In the metformin-treated group, this level decreased compared with the IR group. On the other hand, Engeletin administration significantly decreased the Cas-9 level, bringing it closer to that in the healthy group (Figure 3B). These results are in agreement with those on the Cas-3 level, confirming that Engeletin reduces IR-induced oxidative damage.

#### 3.2.6. TNF-α mRNA Expression Results

TNF-α is a further parameter to examine when considering the damage caused by insulin resistance in the cell. When the expression levels of this molecule were analyzed, significantly higher expression was observed in the IR group compared with the healthy group. The metformin treatment group showed a significant decrease compared with the IR group. On the other hand, Engeletin administration significantly decreased TNF-α levels, bringing them closer to those in the healthy group (Figure 3C). These results support the findings of Cas-9 and Cas-3, further confirming that Engeletin reduces IR-induced oxidative damage.

### 3.3. Biochemical Analyses Results

In our study, the levels of MDA, GSH, and SOD were measured biochemically to examine the increase in oxidative damage due to insulin resistance and the effect of Engeletin.

#### 3.3.1. MDA Analysis Results

When examining the malondialdehyde levels, a significant increase was observed in the IR group compared with the healthy group. Metformin administration significantly decreased the levels of this compound in the IR group compared with the non-IR group. Engeletin administration significantly increased the MDA levels, bringing them closer to those in the healthy group (Figure 4A). These results reveal that Engeletin reduces the oxidative damage due to IR.

#### 3.3.2. GSH Analysis Results

Unlike malondialdehyde, glutathione is a molecule of the intracellular antioxidant system. The presence and amount of this molecule increase the protective level against oxidative damage.

When analyzing the GSH levels in our study, a significant decrease was observed in the IR group compared with the healthy group. In the metformin-treated group, a significant increase in the amount of this compound was measured compared with the IR group. Engeletin administration significantly increased the GSH levels, bringing them closer to those in the healthy group (Figure 4B). These results indicate that Engeletin reduces IR-induced oxidative damage by increasing the GSH levels.

#### 3.3.3. SOD Analysis Results

Superoxide dismutase is an intracellular antioxidant enzyme and a molecule of the intracellular antioxidant system. The presence and amount of this molecule increase the level of protection against oxidative damage.

When analyzing the SOD levels in our study, a significant decrease was observed in the IR group compared with the healthy group. In the metformin-treated group, a significant increase in the levels of this compound was found compared with the IR group. Engeletin administration significantly increased the SOD levels, bringing them closer to those in the healthy group (Figure 4C). These results suggest that Engeletin alleviates the IR-induced oxidative damage by increasing the SOD levels.

## 4. Discussion

Insulin resistance causes metabolic diseases such as obesity, diabetes, hypertension, arteriosclerosis, and polycystic ovary syndrome (PCOS), or it may also occur as a consequence of these diseases. If left untreated, insulin resistance, which is widespread in society, can lead to many diseases ranging from heart disease to infertility. Therefore, many studies have been carried out in recent years to find a solution to insulin resistance and elucidate its mechanism. For a similar purpose, we investigated the efficacy of the active substance Engeletin, which has recently been newly identified in the literature on experimental animals, against insulin resistance in HepG2 human liver cells in vitro.

Insulin resistance is defined as the impaired biological response of target tissues to insulin stimulation. Insulin resistance may affect any tissue that has insulin receptors; however, it is primarily caused by the liver, skeletal muscle, and adipose tissue. Due to this condition, glucose excretion is impaired, resulting in hyperinsulinemia [20,21]. Several factors have been described as the possible reasons underlying the emergence of insulin resistance, such as an increase in visceral adiposity and genetic deformations in one or more proteins involved in the insulin mechanism of action. This often occurs as part of cardiovascular/metabolic diseases called “insulin resistance syndrome” or “metabolic syndrome.” In individuals with insulin resistance, this can lead to type 2 diabetes, polycystic ovary syndrome, non-alcoholic fatty liver disease, microvascular disease, macrovascular disease, and atherosclerosis [22]. IR has become a major source of mortality and morbidity worldwide, as it causes many chronic diseases, such as type 2 diabetes and cardiovascular and cerebrovascular diseases, and has also come to represent a significant cost burden on health systems [23]. The cost burden of IR and metabolic diseases has encouraged the scientific community to study disease mechanisms. It will be critical to unravel disease mechanisms and develop treatment pathways, as IR, obesity, and metabolic diseases are interrelated through molecular–biochemical mechanisms, and the onset of one disease triggers that of the other [24].

Currently, antihyperglycemic medications including metformin, thiazolidinediones (TZDs), and sodium glucose transporter (SGLT)-2 inhibitors are the primary treatments for insulin resistance. Metformin, the most commonly used drug, has been the first-line treatment for type 2 diabetes for many years and has recently been used in the treatment of multiple diseases, including metabolic and cardiovascular diseases [25]. Given that IR is becoming more widespread and is a major trigger of multiple diseases, it will be critical to have sufficient knowledge of its etiology and pathogenesis and develop new diagnostic and therapeutic strategies. Thus, effectively reducing IR and delaying its emergence should become a focus in relevant fields. Many natural remedies, including the use of flavonoid groups, have been studied for this purpose [26]. Flavonoids from various plants have anti-inflammatory and anti-diabetic effects. Several studies have also shown the potential benefits of flavonoids in reducing IR and diabetes [4,27]. Therefore, Engeletin, a flavonoid compound known for its antioxidant and anti-inflammatory pharmacological effects, is a promising candidate for drug development. In this study, we found that the administration of Engeletin alone and in combination with metformin increased glucose consumption by decreasing insulin resistance. Then, we evaluated the mechanism of this effect of Engeletin on insulin resistance based on parameters related to glucose metabolism, such as ISR-1, ISR-2, and GLUT-2.

The insulin receptor substrates ISR-1 and ISR-2 are the main substrates that determine the action of insulin. Insulin signal transduction begins with the integration of insulin into the insulin receptor (IR), which results in the stimulation of several intracellular protein substrates, including ISR-1 and ISR-2 [28]. Based on the analyses performed in this study, the co-administration of metformin and Engeletin increased the expression level of ISR-1, showing a synergistic effect; in particular, the results showed that Engeletin increased the expression of ISR-1 at a dose of 100 µM and showed activity against insulin resistance. The analysis revealed that the co-administration of metformin and Engeletin brought the ISR-2 level closer to the healthy group’s level, and the results showed that Engeletin increased the expression of ISR-2 at a dose of 100 µM. Similarly, in a study using epicatechin and its polyphenolic extract in the literature, the IR/ISR levels in HepG2 cells were investigated, and it was determined that epicatechin and its polyphenolic extract prevented the blockade of insulin signaling and maintained HepG2 functionality by regulating glycogen content [29]. Another study using flavonoid fractions obtained from Glycyrrhiza glabra showed that prenylated flavonoid fractions increased glucose metabolism by inhibiting the activation of the ISR-1 pathway in IR-HepG2 cells [30].

GLUTs are structures that transport glucose across the plasma membrane based on facilitated diffusion. GLUT-2, a type of glucose transporter, is found in liver, kidney, and pancreatic beta cells. Its inactivation in the liver leads to impaired glucose-stimulated insulin secretion. Furthermore, GLUT-2 serves an important role in the control of cellular mechanisms that affect the gene expression of glucose transport activity [31]. GLUT-2 also plays an important role in maintaining homeostasis under hyperglycemic conditions, and the dysregulation of this transporter may lead to major metabolic diseases such as diabetes. Thus, regulating GLUT-2 activity is currently one of the preferred options for diabetes treatment [32]. According to the molecular analyses performed in this study, the GLUT-2 expression levels were significantly decreased in the IR group compared with the healthy group. The results indicated that Engeletin at a dose of 100 µM increased the levels of GLUT-2 by showing metformin-like activation of this transporter. Evidence from various studies has emphasized the important role of flavonoid groups in increasing glucose uptake and the expression of glucose transporter proteins [33]. In a study using ginsenoside against IR in the HepG2 cell line, it was found that GLUT-2 expression was significantly down-regulated in the IR group and significantly increased after ginsenoside application [34]. In this study, we examined the activity of Engeletin on the glucose transporter protein GLUT-2 in order to provide new insights into the potential role of flavonoids.

Another effect of insulin resistance in cells is the stress caused by insufficient glucose reaching the cells. This is also known as oxidative cell damage. The imbalance between the production of reactive oxygen species (ROS) and the capacity of the antioxidant defense system to neutralize ROS leads to cell death, resulting in apoptosis and necrosis. ROS cause insulin resistance in peripheral tissues by affecting insulin receptor signaling pathways [35]. MDA, an indicator of oxidative stress, is known as the end product of lipid peroxidation, while SOD is a free radical scavenger. GSH is responsible for removing reactive oxygen molecules from the cell [36]. In our study, we tested the effect of Engeletin on this damage mechanism. According to our findings, we determined that Engeletin decreased oxidative stress by reducing the elevated MDA levels and increasing the decreased SOD and GSH levels under IR conditions. Huang et al., in a study on erastin-induced oxidative stress in bone marrow mesenchymal stem cells, observed that the administration of Engeletin decreased the MDA levels and increased the SOD and GSH levels [37]. Similarly, in a study using quercetin, another flavonoid, the authors evaluated antioxidant defense enzymes in an HepG2 cell in vitro model and concluded that quercetin reduced oxidative stress [38]. In another study investigating the anti-diabetic effect of flavonoids obtained from Sanguis draxonis in type 2 diabetic rats, it was found in serum that the MDA levels decreased and the SOD and GSH levels increased [39].

Insulin resistance, a state of impaired cellular response to insulin, is closely associated with oxidative stress and inflammation, often culminating in apoptosis. Reduced insulin signaling can disrupt glucose metabolism, leading to hyperglycemia and increased the production of reactive oxygen species (ROS), thereby inducing oxidative stress. This increased oxidative stress, characterized by an imbalance between ROS production and antioxidant defenses, triggers inflammatory pathways by activating transcription factors such as NF-κB. Chronic inflammation, in turn, exacerbates insulin resistance by disrupting insulin receptor signaling and promoting the release of pro-inflammatory cytokines. In addition, persistent oxidative stress and inflammation can overwhelm cellular repair mechanisms, leading to cellular damage and the activation of apoptotic pathways, resulting in programmed cell death in insulin-sensitive tissues such as pancreatic beta cell, muscle, and adipose tissue; and further contributing to metabolic dysfunction and disease progression. An important mediator involved in apoptotic processes is TNF-α. TNF-alpha (TNF-α) is a key cytokine and a potent driver of inflammatory processes in the body. In addition to its role in inflammation, TNF-α can also initiate apoptotic pathways leading to programmed cell death. This dual functionality highlights the importance of TNF-α in the regulation of immune responses and tissue homeostasis, where controlled cell death can resolve inflammation [40]. However, the dysregulation of TNF-α can lead to chronic inflammatory conditions and excessive apoptosis, contributing to various pathological conditions. Caspase-9 is a key initiator of the intrinsic apoptotic pathway that is activated within the apoptosome complex in response to cellular stress signals such as DNA damage or mitochondrial dysfunction. Once activated, caspase-9 proteolytically cleaves and activates downstream executioner caspases, most notably caspase-3 [41]. Activated caspase-3 then induces the dismantling of the cell by cleaving numerous structural and regulatory proteins, leading to the characteristic morphological and biochemical changes associated with apoptosis, which ultimately results in cell death [42]. Studies have shown that TNF-a, Cas-9, and Cas-3 expression increases under insulin resistance conditions [41].

In our study, TNF-a, Cas-9, and Cas-3 expression was significantly increased in the group that developed IR compared with the control group, while it was significantly decreased in the Engeletin groups. Previous studies have demonstrated the anti-inflammatory and anti-apoptotic effects of Engeletin in different experiments, which supports our findings. Li et al. demonstrated that Engeletin reduced intervertebral disk degeneration with its anti-inflammatory and anti-apoptotic effect [43]. Wang et al. also demonstrated that Engeletin provided improvement with anti-inflammatory and anti-apoptotic effects in a TNF-α-induced osteoarthritis model [44].

## 5. Conclusions

This study highlighted the therapeutic effects of Engeletin, a natural flavonoid, on insulin resistance induced in human HepG2 liver cells. Our findings demonstrated that Engeletin treatment was effective on insulin signaling pathways (ISR-1, ISR-2, and GLUT-2) and also regulated biomarkers associated with oxidative stress (MDA, GSH, SOD) and apoptosis (caspase-3, caspase-9, TNF-α). By emphasizing the therapeutic potential of natural compounds, this study is expected to serve as a guiding reference for future scientific research in this field.

## Figures and Tables

**Figure 1 cimb-47-00535-f001:**
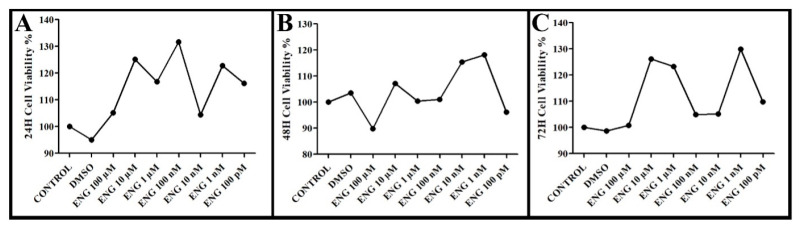
Results of cell viability test (MTT) of Engeletin in HepG2 cell line at (**A**) 24 h, (**B**) at 48 h, and (**C**) at 72 h (DMSO: dimethyl sulfoxide; ENG: Engeletin; µM: Micromolar; nM: Nanomolar; pM: Picomolar).

**Figure 2 cimb-47-00535-f002:**
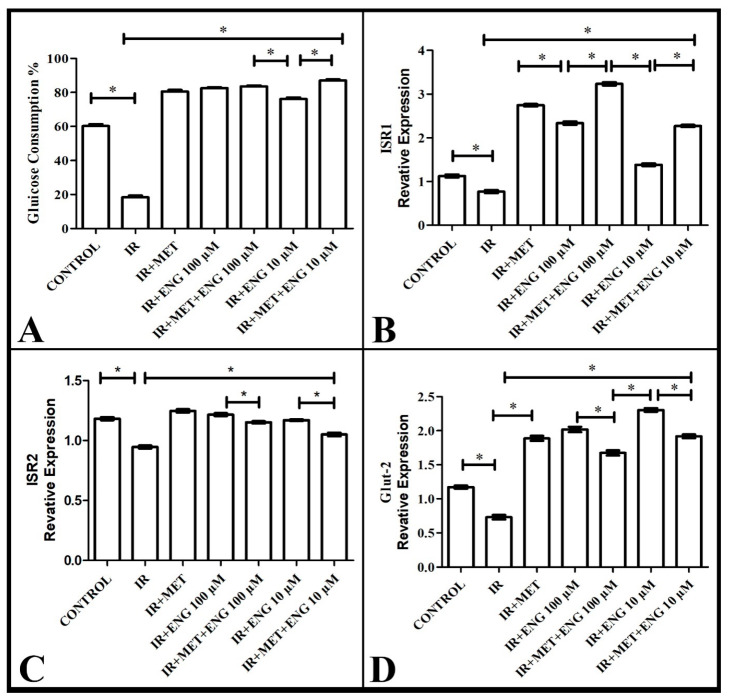
(**A**) The effect of Engeletin on glucose consumption in a model of insulin resistance; (**B**): ISR-1 mRNA expression results; (**C**): ISR-2 mRNA expression results; (**D**): GLUT-2 mRNA expression results (ENG: Engeletin; IR: insulin resistance; MET: metformin). *: *p* < 0.05.

**Figure 3 cimb-47-00535-f003:**
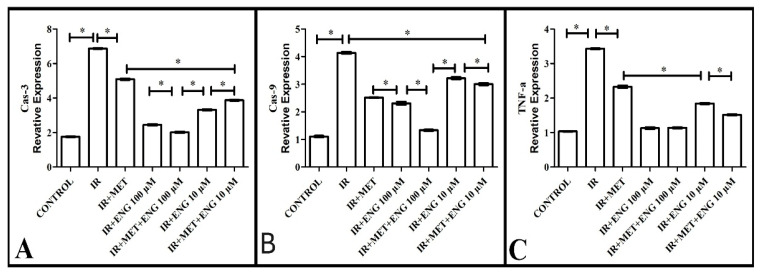
(**A**)**.** The effect of Engeletin on Cas-3 mRNA expression, (**B**) Cas-9 mRNA expression, and (**C**) TNF-α mRNA expression (ENG: Engeletin; IR: insulin resistance; MET: metformin). *: *p* < 0.05.

**Figure 4 cimb-47-00535-f004:**
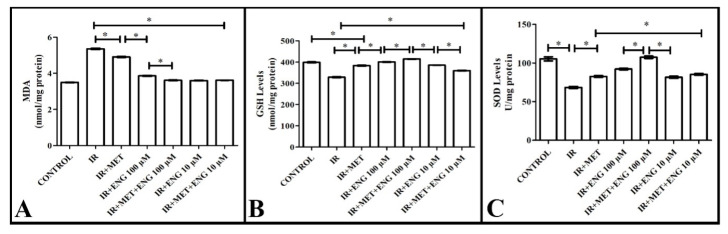
(**A**)**.** The analysis results of the effect of Engeletin on MDA, (**B**) GSH, and (**C**) SOD (ENG: Engeletin; IR: insulin resistance; MET: metformin). *: *p* < 0.05.

**Table 1 cimb-47-00535-t001:** Sequences of primers used for real-time PCR assays.

Human Bactin-F	5′-ATT GCC GAC AGG ATG CAG AAG-3′—21 bp
Human Bactin-R	5′-AGA AGC ATT TGC GGT GGA CG-3′—20 bp
Human Cas9-F	5′-CTT CGT TTC TGC GAA CTA ACA GG-3′—23 bp
Human Cas9-R	5′-GCA CCA CTG GGG TAA GGT TT-3′—20 bp
Human TNF-a-F	5′-CCT CTC TCT AAT CAG CCC TCT G-3′—22 bp
Human TNF-a-R	5′-GAG GAC CTG GGA GTA GAT GAG-3′—21 bp
Human Insulin R1-F	5′-CCC AGG ACC CGC ATT CAA A-3′—19 bp
Human Insulin R1-R	5′-GGC GGT AGA TAC CAA TCA GGT-3′—21 bp
Human Insulin R2-F	5′-ACT TCA CAT TGC AAA CGC CT-3′—20 bp
Human Insulin R2-R	5′-GGA ATT GCT AGC ACG CCT AC-3′—20 bp
Human Glut2-F	5′-GCT GCT CAA CTA ATC ACC ATG C-3′—22 bp
Human Glut2-R	5′-TGG TCC CAA TTT TGA AAA CCC C-3′—22 bp
Human Cas3-F	5′-TGG CGA AAT TCA AAG GAT G-3′—19 bp
Human Cas3-R	5′-TAA CCC GGG TAA GAA TGT GC-3′—20 bp

## Data Availability

The data are contained within the article.

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
