# Peer review of "The Effects of Engeletin on Insulin Resistance Induced in Human HepG2 Liver Cells"

_cimb, 2025, doi:10.3390/cimb47070535_

Round 1
Reviewer 1 Report
Comments and Suggestions for Authors
The manuscript by Topkay et al., is aimed to evaluate the in-vitro effect of Engeletin on insulin resistance and oxidative stress human HepG2 cells. The manuscript is well written and organized. The results are clear and very punctual.
Mayor concerns:
- The effects of Engeletin were investigated in a so-called insulin resistance model in HepG2 cell. However, the information regarding the validation of the model is missing.
- The authors mentioned that they measured insulin-dependent glucose consumption. However, no isotopic neither other methodology (like enzyme activity) to measure consumption was employed. The authors rather measure uptake, but not consumption.
- It is mentioned that “…evaluated the mechanism of this effect of Engeletin on insulin resistance through parameters related to glucose metabolism, such as ISR-1, ISR-2, and GLUT-2…” IRS-1 and 2, are not related to glucose metabolism. Also GLUT-2 is not directly related to glucose metabolism.
- The discussion is extremely long considering the results obtained. At the current format is too speculative.
- The conclusion is not appropriated. It must be re-written omitting all comments regarding conditional future experiments.
Minor concerns:
Fig. 1 A, B and C. ENG 100 uM was written twice on the axis (or something is missing).
The sentence, “Due to resistance, glucose excretion is impaired, resulting in hyperinsulinemia” is difficult to understand in the context of the manuscript. Perhaps glucose handling is more appropriate.
Author Response
|
Response to Reviewer Comments
|
||
|
1. Summary |
|
|
|
We sincerely thank the reviewers for their valuable comments and constructive suggestions, which greatly helped us to improve the quality of our manuscript. We carefully addressed each comment and made the necessary changes in the revised version. Detailed responses to each point raised by the reviewers are provided below. The revisions made to the manuscript have been clearly explained and the changes are indicated with page numbers. We believe that the revised manuscript has been significantly improved and is now suitable for publication.
|
||
|
2. Questions for General Evaluation |
Reviewer’s Evaluation |
Response and Revisions |
|
Does the introduction provide sufficient background and include all relevant references? |
Yes |
The reviewer confirmed that the introduction provides sufficient background and includes relevant references. |
|
Are all the cited references relevant to the research? |
Yes |
All cited references are directly relevant to the research and appropriately support the context of the study. |
|
Is the research design appropriate? |
Must be improved |
The reviewer indicated that the research design needs improvement. Accordingly, the design section has been revised to clarify the methodology and enhance its coherence with the research objectives. |
|
Are the methods adequately described? |
Must be improved |
The methods section has been revised to improve clarity and provide more detailed explanation. |
|
Are the results clearly presented? |
Can be improved |
We sincerely thank the reviewer for the comment. We believe the current presentation of the results clearly conveys the key findings. |
|
Are the conclusions supported by the results? |
Must be improved |
We appreciate the reviewer’s feedback. The conclusion section has been revised to better align with and clearly reflect the presented results. |
|
3. Point-by-point response to Comments and Suggestions for Authors
Comments 1: [The effects of Engeletin were investigated in a so-called insulin resistance model in HepG2 cell. However, the information regarding the validation of the model is missing].
|
||
|
Response 1: Thank you for pointing this out. [Dear reviewer, first of all, thank you for your comments and constructive criticism. When the literature is examined, there are many different applications for the induction of insulin resistance in HepG2 cells. For this reason, we first checked the non-toxicity of glucose at concentrations of 5.5, 16, 20, 25, 30 and 40 mM and insulin at concentrations of 10-5-10-9 M separately or in combination with MTT. Then we evaluated the effects of glucose and insulin separately or in combination on insulin consumption at 24 and 48 hours. After waiting for the optimum concentration and time, glycogen staining test was performed. In this way, we tried to confirm the presence of resistance with 2 different applications in our study. In our project, we would have liked to measure 2-deoxy-D-[14C] glucose uptake if there was sufficient financial opportunity, but we could not realize it].
|
||
|
Comments 2: [The authors mentioned that they measured insulin-dependent glucose consumption. However, no isotopic neither other methodology (like enzyme activity) to measure consumption was employed. The authors rather measure uptake, but not consumption].
|
||
|
Response 2: Thank you for pointing this out. We agree with this comment. [Dear reviewer, you are actually right about what you mentioned, but in most studies in the literature, although the amount of glucose in the supernatant is measured, it is expressed as glucose consumption value. For this reason, we preferred to write it like this].
Comments 3: [It is mentioned that “…evaluated the mechanism of this effect of Engeletin on insulin resistance through parameters related to glucose metabolism, such as ISR-1, ISR-2, and GLUT-2…” IRS-1 and 2, are not related to glucose metabolism. Also GLUT-2 is not directly related to glucose metabolism].
Response 3: Thank you for pointing this out. [Thank you for your valuable comment. We respectfully disagree and would like to clarify our interpretation with supporting references. The insulin receptor substrate proteins ISR-1 and ISR-2 are key targets of the insulin receptor tyrosine kinase and are required for hormonal control of metabolism. Moreover, these proteins have significant effects, particularly on glucose uptake and metabolic homeostasis (Copps and White 2012)]. Similarly GLUT-2 is a glucose transporter predominantly expressed in hepatocytes and pancreatic beta cells, directly involved in glucose uptake and sensing. Nonetheless, the GLUT-2 transporter isoform plays a crucial role in the physiological regulation of glucose-sensitive genes (Thorens 2015)]. Copps, K. D., & White, M. F. (2012). Regulation of insulin sensitivity by serine/threonine phosphorylation of insulin receptor substrate proteins IRS1 and IRS2. Diabetologia, 55(10), 2565–2582. https://doi.org/10.1007/s00125-012-2644-8 Thorens B. (2015). GLUT2, glucose sensing and glucose homeostasis. Diabetologia, 58(2), 221–232. https://doi.org/10.1007/s00125-014-3451-1
Comments 4: [The discussion is extremely long considering the results obtained. At the current format is too speculative].
Response 4: Thank you for pointing this out. [One of our intentions in the discussion section was to provide sufficient background information on the physiological roles of ISR-1, ISR-2 and GLUT-2 in glucose metabolism. These molecules are key components of the insulin signaling pathway, and outlining their mechanisms aimed to help readers-especially those from diverse biomedical fields-clearly understand their relevance in the context of insulin resistance. The detailed explanation was therefore intentional to support the biological plausibility of our findings. For this reason, the discussion section was extended].
Comments 5: [The conclusion is not appropriated. It must be re-written omitting all comments regarding conditional future experiments].
Response 5: Thank you for pointing this out. We agree with this comment. [The conclusions section has been revised, as shown on page 12].
Comments 6: [Fig. 1 A, B and C. ENG 100 uM was written twice on the axis (or something is missing)].
Response 6: Thank you for pointing this out. We agree with this comment. [Figure 1 has been revised, as shown on page 6].
Comments 7: [The sentence, “Due to resistance, glucose excretion is impaired, resulting in hyperinsulinemia” is difficult to understand in the context of the manuscript. Perhaps glucose handling is more appropriate].
Response 7: Thank you for pointing this out. We agree with this comment. [The sentence has been revised as: [“Due to this condition, glucose excretion is impaired, resulting in hyperinsulinemia.”]. As shown on page 9, the sentence has been revised to improve clarity and accuracy.
|
||
|
4.Response to Comments on the Quality of English Language |
||
|
Point 1: The English is fine and does not require any improvement. |
||
|
Response 1: We sincerely thank the reviewer for the positive comment regarding the language quality. Although the manuscript was considered satisfactory, we have nevertheless benefited from MDPI’s professional English editing services to further improve the clarity and overall quality of the text.
|
||
|
5. Additional clarifications |
||
|
[We have no further comments to add. Thank you for your thorough review.] "Please find the attached file for your review." |
||
Reviewer 2 Report
Comments and Suggestions for Authors
The authors investigated the effect of Engeletin on insulin resistance and associated oxidative cell damage in human HepG2 liver cells. This topic is interesting due to the scarcity of literature in this field. However, I identified several critical issues in the manuscript that the authors must address.
The description of experimental procedure has to be improved:
- Authors said "Engeletin was obtained from the supplier company", could the authors please provide the exact name and details of the supplier?
- Concentrations should be expressed using standard units such as mM, µM, nM, pM instead of 10^-4M, 10^-5M, etc
- The use of trypsin to detach cells from the plate surface is a well-known procedure and does not require detailed description.
- Centrifuge speed should be reported in relative centrifugal force (g) rather than rpm.
- Although the authors reproduced a validated experimental protocol to induce insulin resistance, they must verify that the cells have indeed developed resistance before proceeding with further assays.
- What exactly do the authors mean by “One hour before the IR induction, Engeletin and metformin were administered”? Do they mean that Engeletin and metformin were given one hour before starting the insulin resistance induction, or one hour before the IR procedure ended?
- What exactly do the authors mean by"Healthy (HEALTHY GROUP)"? Do they mean HepG2 liver cells without insulin resistance induced?
- Authors said "Total RNA isolation steps were
carried out as recommended by the manufacturer", could the authors please provide the exact name and details of the manufacturer?
- Authors said "Total mRNA was obtained", could the authors specify if they provided a quantification of isolated RNA? The quantification of RNA is mandatory for the cDNA synthesis and for Real-Time PCR assay
- The authors should provide the sequences of the primers used for the Real-Time PCR assays
- In section 2.5.2, the authors state “according to the following temperature values,” but no temperature details or protocols are provided
- In section 2.5.3, the authors mention that “SYBR Green was pipetted according to the recommended amounts of the company,” but no further details or specifications are provided about composition of the mixture used for the Real-time PCR assay
- In section 2.6.1. authors state that used "human-specific ELISA kits to measure malondialdehyde (MDA), glutathione (GSH), and superoxide dismutase (SOD) levels following the procedures specified in the kit procedures" without providing any details about the company furnished the reagents
- Statistically significant values should be highlighted in the graphs using asterisks (*) (Fig. 1), as is standard practice, rather than using combinations of letters, as currently seen in Figures 2, 3, and 4. Please modify the figures.
- Authors reported the mRNA level of investigated genes as concentration [pg/ml]. Could the authors please clarify the method they used to determine the concentration of mRNA?
- The standard deviation is almost completely absent in all the data presented in the graphs. Could the authors please provide the raw data obtained from the instruments, both from the real-time PCR and the plate reader?
- The English language of the manuscript should be improved with the assistance of specialized editing services
The English language of the manuscript should be improved with the assistance of specialized editing services
Author Response
|
Response to Reviewer Comments
|
||||||||||||||||||||||||||||||
|
1. Summary |
|
|
||||||||||||||||||||||||||||
|
We sincerely thank you for taking the time to review our manuscript and for providing thoughtful and constructive feedback. All comments and suggestions have been carefully evaluated and addressed in the revised version of the manuscript. The changes made are clearly highlighted and explained in detail below. |
||||||||||||||||||||||||||||||
|
2. Questions for General Evaluation |
Reviewer’s Evaluation |
Response and Revisions |
||||||||||||||||||||||||||||
|
Does the introduction provide sufficient background and include all relevant references? |
Yes |
Thank you for your positive comment. We are glad that the introduction section was found to provide sufficient background information. |
||||||||||||||||||||||||||||
|
Are all the cited references relevant to the research? |
Yes |
Thank you for confirming that all references are relevant to the study. |
||||||||||||||||||||||||||||
|
Is the research design appropriate? |
Must be improved |
We thank the reviewer for the constructive comment regarding the study design. In response, we have revised the relevant sections as suggested. The improvements made to the study design are detailed below with corresponding page numbers. |
||||||||||||||||||||||||||||
|
Are the methods adequately described? |
Must be improved |
Thank you for your feedback. The methods section has been revised in accordance with the reviewer’s suggestions. |
||||||||||||||||||||||||||||
|
Are the results clearly presented? |
Must be improved |
Thank you for your comment. To improve the clarity of the results, we have revised the figures in the results section accordingly. |
||||||||||||||||||||||||||||
|
Are the conclusions supported by the results? |
Yes |
Thank you for your positive comments. We are pleased that the results were found to support the conclusions. |
||||||||||||||||||||||||||||
|
3. Point-by-point response to Comments and Suggestions for Authors |
||||||||||||||||||||||||||||||
|
Comments 1: [ Authors said "Engeletin was obtained from the supplier company", could the authors please provide the exact name and details of the supplier?]
|
||||||||||||||||||||||||||||||
|
Response 1: Thank you for pointing this out. We agree with this comment. The requested section (2.2) has been revised, as shown on page 3 of the manuscript. [Engeletin was obtained from the supplier company (Medchemexpress company, Engeletin: HY-N0436)]
|
||||||||||||||||||||||||||||||
|
Comments 2: [ Concentrations should be expressed using standard units such as mM, µM, nM, pM instead of 10^-4M, 10^-5M, etc]
|
||||||||||||||||||||||||||||||
|
Response 2: Thank you for pointing this out. We agree with this comment. The requested section (2.2) has been revised, as shown on page 3 of the manuscript. [We changed as 100 µM, 10 µM, 1 µM, 100 nM, 10 nM, 1 nM, and 100 pM in whole manuscript.]
Comments 3: [The use of trypsin to detach cells from the plate surface is a well-known procedure and does not require detailed description.]
Response 3: Thank you for pointing this out. We agree with this comment. [The requested modification is presented in section 2.2 and can be seen on page 3 of the manuscript. In accordance with the reviewer’s suggestion, the detailed explanation regarding trypsin has been removed to enhance the clarity and focus of the manuscript.]
Comments 4: [Centrifuge speed should be reported in relative centrifugal force (g) rather than rpm.]
Response 4: Thank you for pointing this out. We agree with this comment. [The requested modification is presented in section 2.2 and can be seen on page 3 of the manuscript. The term “rpm,” which was mentioned in the context of the trypsinization procedure, has been removed as per the reviewer’s comment.]
Comments 5: [Although the authors reproduced a validated experimental protocol to induce insulin resistance, they must verify that the cells have indeed developed resistance before proceeding with further assays.]
Response 5: Thank you for pointing this out. [Dear reviewer, similarly to the criticisms of the previous reviewer, I would like to express this situation to you in the following way. When the literature is examined, there are many different applications for the induction of insulin resistance in hepg2 cells. For this reason, we first checked the non-toxicity of glucose at concentrations of 5.5, 16, 20, 25, 30 and 40 mM and insulin at concentrations of 10-5-10-9 M separately or in combination with MTT. Then we evaluated the effects of glucose and insulin separately or in combination on insulin consumption at 24 and 48 hours. After waiting for the optimum concentration and time, glycogen staining test was performed. In this way, we tried to confirm the presence of resistance with 2 different applications in our study. In our project, we would have liked to measure 2-deoxy-D-[14C] glucose uptake if there was sufficient financial opportunity, but we could not realize it.]
Comments 6: [What exactly do the authors mean by “One hour before the IR induction, Engeletin and metformin were administered”? Do they mean that Engeletin and metformin were given one hour before starting the insulin resistance induction, or one hour before the IR procedure ended?]
Response 6: Thank you for pointing this out. [As you mentioned, dear reviewer, it was given 1 hour before inducing insulin resistance. It was written more clearly in the article as you suggested.]
Comments 7: [What exactly do the authors mean by"Healthy (HEALTHY GROUP)"? Do they mean HepG2 liver cells without insulin resistance induced?]
Response 7: Thank you for pointing this out. [Thank you for the question. The requested modification is presented in section 2.4 and can be seen on page 3 of the manuscript. To clarify, we have defined the healthy group as the control group, and it is now explicitly stated in the manuscript as “control group (healthy group)”.]
Comments 8: [Authors said "Total RNA isolation steps were carried out as recommended by the manufacturer", could the authors please provide the exact name and details of the manufacturer?]
Response 8: Thank you for pointing this out. The requested modification has been revised in section 2.5.1 and can be seen on page 4 of the manuscript. [Total RNA isolation steps were carried out as recommended by the manufacturer. Total mRNA was obtained. Total mRNA was purified using the RNeasy Mini Kit on the QIACUBE (Qiagen, Hilden, Germany) device according to the manufacturer’s instructions.]
Comments 9: [Authors said "Total mRNA was obtained", could the authors specify if they provided a quantification of isolated RNA? The quantification of RNA is mandatory for the cDNA synthesis and for Real-Time PCR assay.]
Response 9: Thank you for pointing this out. The requested modification has been revised in section 2.5.1 and can be seen on page 4 of the manuscript. [Dear reviewer, Total mRNA amount was measured by nano drop spectrophotometry (EPOCH Take3 Plate, Biotek) at 260 nm.]
Comments 10: [The authors should provide the sequences of the primers used for the Real-Time PCR assays]
Response 10: Thank you for pointing this out. We agree with this comment. [The requested modification can be seen in section 2.5.3 on page 5 of the manuscript.] Table 1. Sequences of primers used for real-time PCR assays.
Comments 11: [In section 2.5.2, the authors state “according to the following temperature values,” but no temperature details or protocols are provided.]
Response 11: Thank you for pointing this out. We agree with this comment. The requested modification has been revised in section 2.5.2 and can be seen on page 4 of the manuscript. [Each reaction was performed with 10μl of RNA, and cDNA synthesis was carried out using the Veriti 96-Well Thermal Cycler (Applied Biosystems) at 25°C for 10 minutes, 37°C for 120 minutes, and 85°C for 5 minutes.]
Comments 12: [In section 2.5.3, the authors mention that “SYBR Green was pipetted according to the recommended amounts of the company,” but no further details or specifications are provided about composition of the mixture used for the Real-time PCR assay.]
Response 12: Thank you for pointing this out. We agree with this comment. The requested modification has been revised in section 2.5.3 and can be seen on page 4 of the manuscript. [mRNA expression was quantified using the instructions of the Applied biosystems PowerUp SYBR Green Master Mix kit. For 10 ng cDNA, PowerUp SYBR Green Master Mix (2X) was pipetted 10 µL and Forward and reverse primers + DNA template + Nuclease-Free Water pipetted 10 µl to a total volume of 20 µl according to the SYBR Green Master Mix user guide, and 40 cycles of PCR were continued according to the following temperature values (if primary Tm <60°C, Standard cycling mode at 2 min 50°C, 2 min 95°C, 15 sec 95°C, 15 sec 55°C, 1 min 72°C)]
Comments 13: [In section 2.6.1. authors state that used "human-specific ELISA kits to measure malondialdehyde (MDA), glutathione (GSH), and superoxide dismutase (SOD) levels following the procedures specified in the kit procedures" without providing any details about the company furnished the reagents]
Response 13: Thank you for pointing this out. We agree with this comment. The requested modification has been revised in section 2.6.1 and can be seen on page 5 of the manuscript. [Oxidant and antioxidant parameters—malondialdehyde (MDA), glutathione (GSH), and superoxide dismutase (SOD) levels—were measured using human-specific ELISA kits (MDA: ELABSCIENCE (E-BC-K025-M); GSH: ELABSCIENCE(E-EL-H5410); and SOD: ELABSCIENCE (E-EL-H1113)) with two repetitions. Protein amounts in the supernatants were measured manually using the Lowry method. The mean absorbance of each sample and standard was calculated. All data were shown as mean ± standard deviation (SD) relative to each mg protein.]
Comments 14: [Statistically significant values should be highlighted in the graphs using asterisks (*) (Fig. 1), as is standard practice, rather than using combinations of letters, as currently seen in Figures 2, 3, and 4. Please modify the figures.]
Response 14: Thank you for pointing this out. We agree with this comment. [Figures 2,3, and 4 have been revised as requested. The corresponding modifications can be seen on pages 7,8, and 9 of the manuscript.]
Comments 15: [Authors reported the mRNA level of investigated genes as concentration [pg/ml]. Could the authors please clarify the method they used to determine the concentration of mRNA?]
Response 15: Thank you for pointing this out. We agree with this comment. The requested modification has been revised in section 2.5.3 and can be seen on page 4 of the manuscript. [It was written as pg/ml by mistake. We express mRNA expressions as fold change by comparing them with the control group. All data were expressed as fold change in expressions compared to the control (healthy) group using the 2−ΔΔCt method (Livak & Schmittgen 2001).] LIVAK KJ & SCHMITTGEN TD. 2001. Analysis of relative gene expression data using real-time quantitative PCR and the 2(-Delta Delta C(T)) Method. Methods 25(4): 402-408.
Comments 16: [The standard deviation is almost completely absent in all the data presented in the graphs. Could the authors please provide the raw data obtained from the instruments, both from the real-time PCR and the plate reader?]
Response 16: Thank you for pointing this out. We agree with this comment. [Dear reviewer, the cell culture experimental processes of this study were carried out in the laboratories of the Kafkas University. However, biochemical and molecular analyses were carried out in the laboratories of Ağrı University and Atatürk University, respectively. Data were analyzed in a double-blind manner. When the institutions were contacted for raw data, it was stated that after the analyzes carried out in 2021, the devices in the central laboratories were moved to the hospitals and used during the pandemic period in order to eliminate the difficulties and disruptions that developed due to the pandemic, especially in hospitals. They stated that they could not help with the raw data of our study.]
Comments 17: [The English language of the manuscript should be improved with the assistance of specialized editing services]
Response 17: Thank you for pointing this out. We agree with this comment. [The language of the manuscript has been edited using MDPI’s professional English editing services to ensure clarity and linguistic accuracy.]
|
||||||||||||||||||||||||||||||
|
|
||||||||||||||||||||||||||||||
Round 2
Reviewer 2 Report
Comments and Suggestions for Authors
Paper can be acceptedin the present form
Author Response
The revisions requested in the second round have been comprehensively addressed in the "Response to Academic Editors" section. In addition, the revised manuscript and relevant supplementary file have been duly uploaded.